# Benchmarking Unsupervised Representation Learning for Continuous Control

Rika Antonova
EECS, KTH, Stockholm, Sweden

Sam Devlin
Microsoft Research

Katja Hofmann
Microsoft Research

Danica Kragic
EECS, KTH, Stockholm, Sweden

*Abstract*—We address the problem of learning reusable state representations from a non-stationary stream of high-dimensional observations. This is important for areas that employ Reinforcement Learning (RL), which yields non-stationary data distributions during training. Unsupervised approaches can be trained on such data streams to produce low-dimensional latent embeddings, which could be reused on domains with different dynamics and rewards. However, there is a need to adequately evaluate the quality of the resulting representations. We propose an evaluation suite that measures alignment between the learned latent states and the true low-dimensional states. Using this suite, we benchmark several widely used unsupervised learning approaches. This uncovers the strengths and limitations of existing approaches that impose additional constraints/assumptions on the latent space.

## I. INTRODUCTION

In recent years, significant attention has been devoted to reinforcement learning (RL) [1], which could be promising for the field of robotics [2]. However, re-training control policies for each task from scratch is prohibitively expensive in most cases. Often, it is simply impossible due to lack of hardware data for each robot+task instance and lack of data-efficiency, e.g. for model-free RL. This problem is especially profound for the case of high-dimensional observations (e.g. RGB images, point clouds). Hence, there is a strong demand for approaches that can learn re-usable low-dimensional representations, which can transfer across different tasks and robot/object dynamics. Consider the case when RL is trained on source domains with ample data (e.g. in simulation or in a setting where exploration is cheap and safe). State representations could be extracted from intermediate layers of RL networks, but they might not be reusable on a target domain with different rewards or dynamics.

The field of unsupervised learning could offer potential solutions for streamline learning of reusable latent embeddings, e.g. with bottleneck-reconstruction approaches, such as variational autoencoder (VAE) [3] variants. However, evaluation in this field has mostly focused on dataset-oriented learning, making a limiting assumption that the training data distribution is stationary. Moreover, advanced unsupervised learning works mostly report best-case results. These may be achievable only with architectures, hyperparameters and learning rate decay that the authors find to work best for a given dataset. Furthermore, obtaining reconstructions that are clear enough to judge whether all the important information is encoded in the latent state could still require days or weeks of training [4, 5]. Hence, there is a need for thorough evaluation that does not rely on simply looking at the reconstructed images.

We introduce an evaluation suite for measuring the quality of unsupervised representation learning for continuous control domains. We extend commonly used benchmarks, implemented using PyBullet [6], to report both low- and high-dimensional state. Low-dimensional state contains standard representations used in robotics, e.g. robot joint angles & velocities. High-dimensional state is expressed by an RGB image of the scene. We provide tools to measure alignment between the latent state from unsupervised learners and the true low-dimensional simulator state. Furthermore, we introduce new environments for manipulation with multiple objects and ability to vary their complexity: from geometric shapes to mesh scans and visualizations of real objects. We analyze several commonly used unsupervised approaches with the proposed evaluation suite. Our experiments show that while alignment with true low-dimensional state is achieved on the simpler benchmarks, the more advanced environments present a formidable challenge, especially for approaches that need to employ reconstruction.

## II. RELATED WORK

Scalable simulation suites for continuous control, such as [7, 8, 9], have the potential to improve applicability of deep RL to robotics. However, advanced benchmarks for unsupervised learning from non-stationary data are lacking, since that community adopted mainly dataset-oriented evaluation. [10] provides such framework for ATARI games, but it is not aimed for continuous control. [11] includes a limited set of robotics domains and 3 metrics for measuring representation quality: KNN-based, correlation, RL reward. We incorporate more standard benchmarks in our suite, introduce a variety of objects with realistic appearances and measure alignment to latent state in a complimentary way: highly non-linear, but not RL-based. In future work, it could be useful to create a combined suite to support both games- and robotics-oriented domains, and offer a comprehensive set of RL-based and RL-free evaluation. In our work, we aimed to create a setting where it is tractable to train & evaluate using a single GPU (or a small number of GPUs/CPUs). This makes our suite applicable to initial evaluation of new algorithms and training adjustments. It could also be beneficial to include more computationally demanding but completely photo-realistic simulated environments, e.g. [12, 13]. Such combined framework could offer more direct incentives for computer vision and learning communities to thoroughly evaluate their proposed approaches on non-stationary data streams, which would be more relevant to robotics.

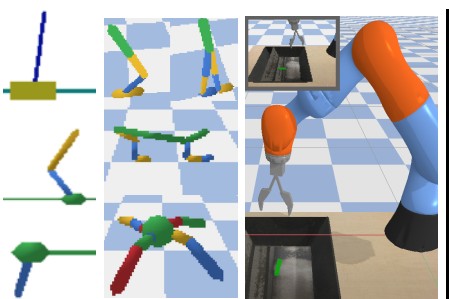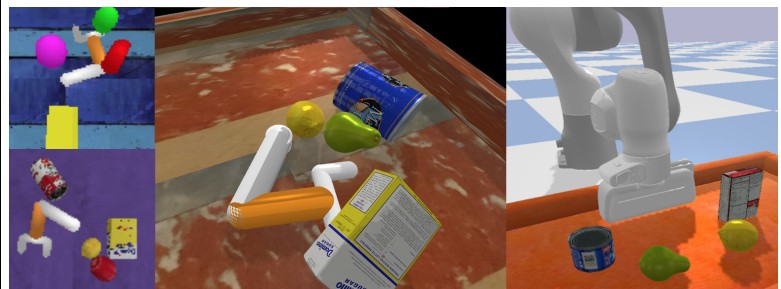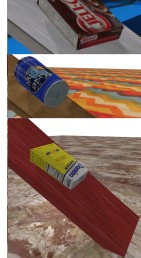

Fig. 1: Evaluation suite environments. Left: Standard PyBullet environments for which our suite yields both pixels and low-dimensional state. Right: Proposed new domains with YCB objects. Our suite is available at https://github.com/contactrika/bulb

## III. EVALUATION SUITE FOR UNSUPERVISED LEARNING

Our proposed evaluation suite tracks the alignment between the learned latent state and the true low-dimensional state. We train unsupervised approaches on frames that an RL policy yields during its own training. This creates a setting with a non-stationary stream of RGB images as training data for an unsupervised learner. The alignment of the learned latent state and the true low-dimensional state is measured periodically as training proceeds. For this, we do a regression fit using a small fully-connected neural network, which takes latents as inputs and is trained to produce low-dimensional states as outputs (robot joint angles, velocities, feet contacts for locomotion, object position/orientation for manipulation). The quality of alignment is characterized by the resulting test error rate. This approach helps quantify latent space quality without the need to wait for clear image reconstructions to emerge. It also provides a general way of judging whether the latent representation is amenable to be incorporated as a low-dimensional state into a larger learning system. This is in contrast to other approaches that measure specific aspects of state quality, such as disentanglement. Our regression-based approach is more general: if learning using the latent state succeeds, disentanglement between dimensions is not strictly needed, and mandating it could limit the flexibility of learned representations.

To connect our suite to existing RL benchmarks, we extend OpenAI gym interface [14] of several widely used robotics domains, so that both pixel- and low-dimensional state are reported during training. We use PyBullet [6] version of these domains, which is open source and free, hence supports wide accessibility/affordability of our testing suite. Simulation environments are parallelized, ensuring a scalable setup. PyBullet provides a convenient python interface, and ensures simulations are fast: its underlying physics engine runs in C/C++. To create more realistic object appearances and dynamics we introduce advanced domains utilizing meshes from scans of real objects from YCB dataset [15]. Our RearrangeYCB domain models object rearrangement tasks, with variants for using a basic planar robot arm and a realistic option with Franka Emika robot arm. RearrangeGeom variant offers an option with simple geometric shapes instead of object scans. YCB-on-incline domain models objects sliding down an incline, with options to change friction and apply external forces; Geom-

on-incline offers a variant with simple single-color geometric shapes. Figure 1 gives an overview; other domains with OpenAI gym interface can be easily incorporated into the suite as well.

## IV. BENCHMARKING LATENT STATE ALIGNMENT

We evaluated several widely used and recently proposed unsupervised learning approaches. Below we give a brief overview of each approach:

– $VAE_{v_0}$ [3]: a VAE with a 4-layer convolutional encoder and corresponding de-convolutional decoder (same conv-deconv stack is also used for all the other VAE-based methods below).

– $VAE_{rpl}$: a VAE with a replay buffer that retains 50% of initial frames from the beginning of training and replays them throughout training. This is our modification of the basic VAE to ensure consistent performance on frames coming from a wider range of RL policies.

– $\beta$-$VAE$ [17]: a VAE with an additional $\beta$ parameter in the variational objective that is supposed to encourage disentanglement of the latent state. To give $\beta$-$VAE$ its best chance we experimented with several $\beta$ parameters and also included replay enhancement from $VAE_{rpl}$.

– $SVAE$: a sequential VAE that is trained to reconstruct a sequence of frames $x_1, ..., x_t$ and passes the output of the convolutional stack through LSTM layer before decoding. Reconstructions for this and other sequential versions were also conditioned on actions $a_1, ..., a_t$.

– $PRED$: a VAE that is given a sequence of frames $x_1, ..., x_t$ and is tasked with constructing a predictive sequence $x_1, ..., x_{t+k}$. First, the convolutional stack is applied to each $x_i$ as before; then, the $t$ output parts are aggregated and passed through several fully connected layers. Their output constitutes the predictive latent state. To decode: this latent code is chunked into $t + k$ parts, each fed into deconv stack for reconstruction.

– $DSA$ [18]: a sequential autoencoder that uses structured variational inference to encourage separation of static vs dynamic aspects of the latent state. It uses LSTMs in static and dynamic encoders. To give $DSA$ its best chance we tried uni- and bidirectional LSTMs, as well as replacing LSTMs with GRUs, RNNs, convolutions and fully connected layers.

– $SPAIR$ [19]: a spatially invariant and faster version of AIR [20] that imposes a particular structure on the latent state. $SPAIR$ overlays a grid over the image (e.g. 4x4=16, 6x6=36 cells) and learns 'location' variables that encode bounding

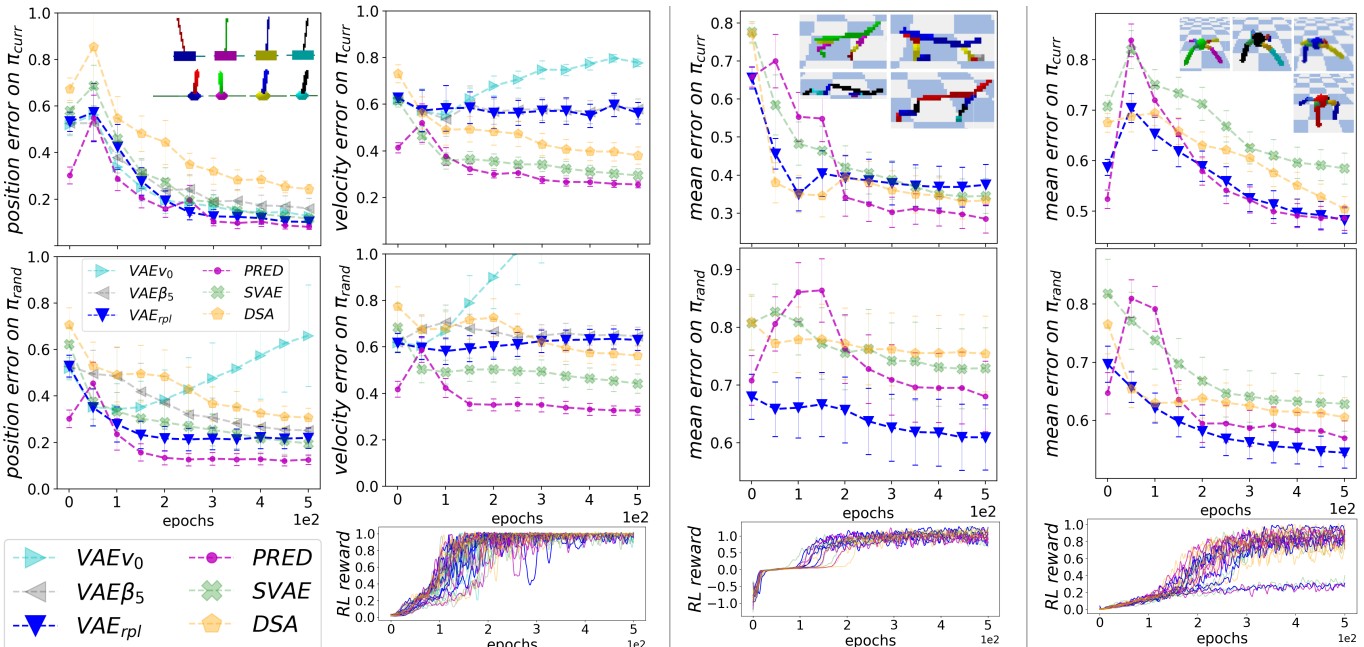

Fig. 2: Benchmarking alignment with true low-dimensional sate on multicolor versions of CartPole, InvertedPendulum, HalfCheetah, Ant. The plots show mean absolute test error of NN regressors trained with current latent codes as inputs and true low-dimensional states (robot positions, velocities, contacts) as outputs. 90% confidence intervals over 6 training runs for each unsupervised approach are shown (overall, we performed >140 training runs for these plots). Unsupervised approaches are trained on frames from replay buffers, filled by PPO [16] RL learner, while it trains from 64 parallel simulation environments. Unsupervised learners get 1024 frames per batch; 10 batches per epoch for pendulums, 50 for locomotion. Top row shows performance on frames from current RL policy $\pi_{curr}$, middle row: random policy $\pi_{rand}$. Current RL reward is displayed in the bottom row (scaled to $\approx$[-1, 1]). 1st column shows results for CartPole and InvertedPendulum for position (cart/base position, pole/pendulum angle); 2nd column: for velocity (cart/base linear velocity, pole/pendulum angular velocity). 3rd column shows aggregated results for position, velocity and contacts for HalfCheetah; 4th column shows these results for the Ant domain.

boxes of objects detected in each cell. 'Presence' variables indicate object presence in a particular cell. A convolutional backbone first extracts features from the overall image (e.g. 64x64 pixels). These are passed on to further processing to learn 'location','presence' and 'appearance' of the objects. The 'appearance' is learned by object encoder-decoder, which only sees a smaller region of the image (e.g. 28x28 pixels) with a single (presumed) object. The object decoder also outputs transparency alphas, which allow rendering occlusions.

### A. Neural Network Architectures and Training Parameters

In our experiments, unsupervised approaches learn from 64x64 pixel images, which are rendered by the simulator. All approaches (except $SPAIR$) first apply a convolutional stack with 4 hidden layers, (with [64,64,128,256] conv filters). The decoder has analogous de-convolutions. Fully-connected and recurrent layers have size 512. We also experimented with batch/weight normalization and larger/smaller network depth & layer sizes, but these did not yield a noticeable change in performance. The latent space size is set to be twice the dimensionality of the true low-dimensional state. For VAE we also tried setting it to be the same, but this did not impact results. $PRED, SVAE, DSA$ use sequence length 24 for pendulums & 16 for locomotion (increasing to 32 yields similar results). $SPAIR$ parameters and network sizes are set to match those

in [19]. We experimented with several alternatives, but only the cell size had a noticeable effect on the final outcome. We report results for 4x4 and 6x6 cell grids, which did best.

To decouple the number of gradient updates for unsupervised learners from the simulator speed: frames for training are re-sampled from replay buffers. These keep 5K frames and replace a random subset with new observations collected from 64 parallel simulation environments, using the current policy of an RL learner. All training hyperparameters are the same for all settings (e.g. learning rate set to 1e-4). Since different approaches need different time to perform gradient updates, we equalize the resources consumed by each approach by reducing the batch size for the more advanced/expensive learners. $VAE_{v_0}, VAE_{rpl}, \beta$-$VAE$ get 1024 frames per batch; for sequential approaches ($SVAE, PRED, DSA$) we divide that by the sequence length; for $SPAIR$ we use 64 frames per batch (since $SPAIR$'s decoding process is significantly more expensive). With that, the compute time of these approaches is roughly equalized.

### B. Evaluation on Multicolor Pendulums and Locomotion

Figure 2 shows results on multicolor versions of CartPole, InvertedPendulum, HalfCheetah and Ant domains. We developed these versions to give a chance to the more advanced algorithms to display their benefits. A potential issue with

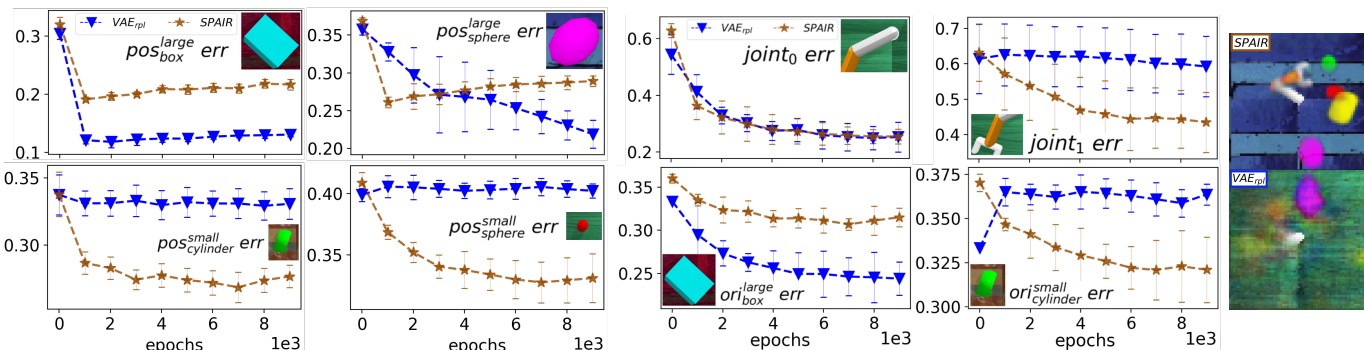

Fig. 3: Evaluation on RearrangeGeom domain. $VAE_{rpl}$ encoded angle of the main robot joint, location & partly orientation (major axis) of the largest objects. $SPAIR$ encoded (rough) locations quickly, but did not improve with longer training.

original environments could have been that the single color scheme in the images was too simple to exploit for obtaining trivial color-based features. With this foresight, we developed the multicolor versions. However, in the end we did not observe qualitatively different results on multicolor vs original domains.

We performed evaluation on frames from two kinds of policies: current RL learner policy $\pi_{curr}$ and random policy $\pi_{rand}$. Ensuring reasonable performance on $\pi_{rand}$ is needed for successful transfer: the stream of frames generated when starting to learn a new task would be more similar to that from a random policy than a final source task policy. Frames used for evaluation were held out from training i.e. not added to replay buffers at any time. To decouple unsupervised learner performance from RL training: for these experiments RL was trained on simulator state. Our evaluation suite supports training RL from the current latent representation. This is useful when analyzing one unsupervised method, but when comparing different methods this would likely cause RL learners to learn in different ways and at different rates. In turn, this could cause evaluation on $\pi_{curr}$ to refer to incomparable policies with vastly different success rates. For example: if RL gets stuck it might produce the same failed end state often, which would be easy to reconstruct due to low variability in frames, but would not constitute a success of the unsupervised learner.

The performance of $VAE_{v_0}$ quickly deteriorated on $\pi_{rand}$. We discovered that this problem can be effectively eliminated by replaying the frames from the initial random policy. The resulting $VAE_{rpl}$ offered good alignment for position-based part of the true state (e.g. cart/base $x$ coordinate, angles of the pole/pendulum, joint angles for HalfCheetah and Ant robots). Hence, we added this fix to all the other approaches as well.

Surprisingly, $\beta$-$VAE$ offered no improvement over $VAE_{rpl}$. We tried a range of $\beta$s: $[100, 20, 10, 5, 0.5]$; the best ($\beta = 5$) performed slightly worse than $VAE_{rpl}$ on pendulum domains (shown in Figure 2), the rest did significantly worse (omitted from plots).

As anticipated, the sequential approaches ($SVAE$, $PRED$, $DSA$) offered significant gains when measuring alignment to velocity part of the true low-dimensional state. Despite its simpler architecture and training, $PRED$ performed best on pendulum domains. For aggregated performance on position, velocity and contacts (i.e. whether robot joints are touching the ground): in locomotion domains $PRED$ outperformed $VAE_{rpl}$ on $\pi_{curr}$, but was second-best on $\pi_{rand}$. Overall, this set of experiments was rather illuminating: simpler approaches were frequently able to beat the more advanced ones, despite the fact we made an extra effort to ensure the appearance of the domains was not trivialized.

### C. Evaluation on the New Multi-object Environments

For our newly proposed domains with multiple objects: the first surprising result was that all approaches we tested failed to achieve clear reconstructions for objects from the YCB dataset. This was despite our attempts of using larger architectures (up to 8 layers with skip connections) to get a decoder network similar to [21]. Clear reconstructions were achieved on the simplified version with geometric shapes with $SPAIR$, while the rest of the algorithms failed to reconstruct the simplified objects as well. This indicates that a multi-object domain (with realistic textures) is a highly needed addition to the current continuous control benchmarks. While single-object benchmarks might be still challenging for control, they could be

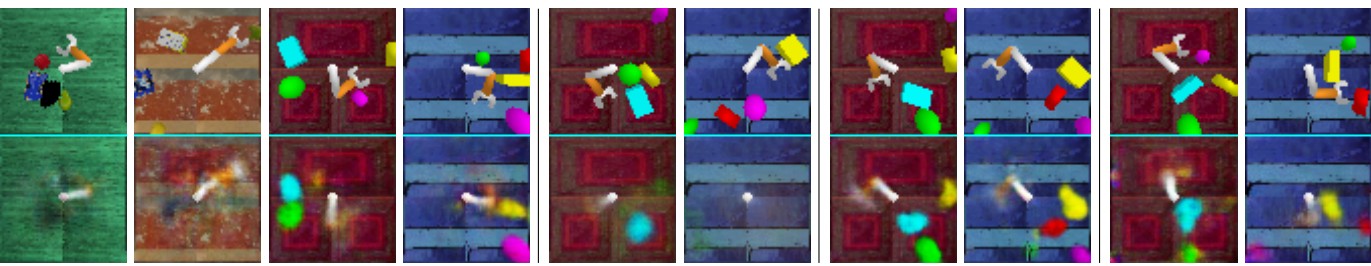

Fig. 4: True images (top) and reconstructions (bottom) after 10K epochs for: $VAE_{rpl}$ (4 leftmost columns), $SVAE$ (next 2 columns), $PRED$ (next 2 columns), $DSA$ (2 rightmost columns).

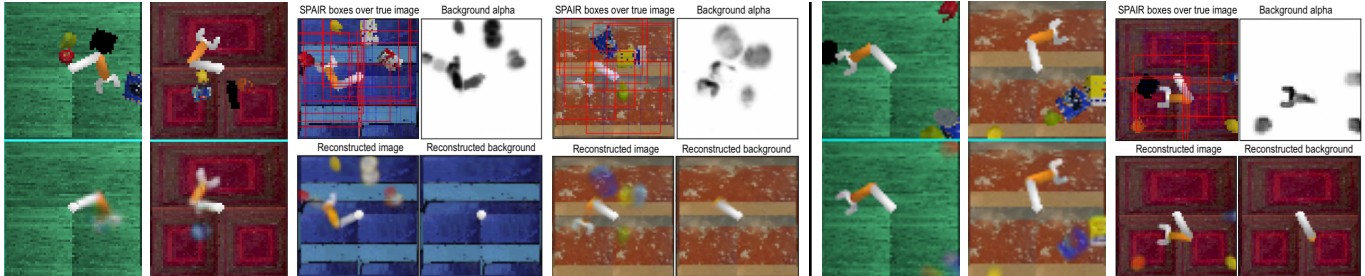

Fig. 5: Left side: SPAIR RearrangeYCB results after 10K epochs ($\approx 32$ hours). Right side: SPAIR after 100K epochs ($\approx 11$ days). True images are in the top row, reconstructions in the bottom. 3 left sets of images show SPAIR with 6x6 grid; 4 right sets show SPAIR with 4x4 grid. Red boxes overlaid over true image show that bounding boxes did not shrink with further training. SPAIR 6x6 tended to split large objects into pieces (visible in the case with blue background). SPAIR 4x4 did not split objects and had better results for low-dimensional alignment. Despite shortcomings, SPAIR's reconstructions were better than those from other approaches we tried. Its object localization could be suitable for cases where approximate positions are sufficient.

inherently simpler for latent state learning and reconstruction. We note that our evaluation uncovered shortcomings of recent approaches, which are considered successful by the learning community. This hints at the fact that it could be challenging to scale/adapt the progress in this other community to make it adequately applicable to advanced applications in robotics. We will come back to this discussion in the conclusions section.

Figure 3 shows analysis of latent state alignment for $SPAIR$ vs $VAE_{rpl}$ on RearrangeGeom domain. $VAE_{rpl}$ only learns to encode position & orientation of the largest object and the angle of the main robot joint. $SPAIR$ encodes positions of all objects quickly, but the alignment remains imprecise even after extensive further training. This could be due to the fact that our benchmark contains objects of various sizes, while $SPAIR$'s most successful results have been shown on domains with small uniformly sized objects. Bounding boxes reported by $SPAIR$ were not tight event after further training (up to 11 days of training on one NVIDIA GeForce GTX1080 GPU). We used PyTorch implementation from [22], which was tested in [23] to reproduce the original $SPAIR$ results (and we added the capability to learn non-trivial backgrounds). An optimized Tensorflow implementation could potentially offer a speedup, but PyTorch has a strong advantage of being more accessible and convenient for research code.

We note that $VAE_{rpl}$ outperformed $SPAIR$ on encoding the orientation of the largest object. This exposes the limitation of $SPAIR$ as position-oriented approach, which succeeds in adding more structure to help encode location information, but does not alleviate the challenge of uncovering orientation information. Orientation has to be inferred from the object 'appearance' features, which lie in the unstructured part of the latent space. Our analysis illustrates that structuring latent space could be beneficial, but has to be done such that it does not impair the learning process and latent representations. This is not trivial, since relying on intuition can be misleading. Seemingly beneficial structure and assumptions that have been shown to work well on simpler domains could fail to hold on a new set of domains in unanticipated ways. At the same time, forgoing structure can result in complete failure to capture information from the more advanced scenes.

It would be interesting to compare SPAIR to a few successors, for example SPACE [24] and IODINE [4]. However, the source code for these was not available. Since these approaches aim to be more general and incorporate less structure than SPAIR: there is no definite reason for them to yield better results in our setting. Nonetheless, it would be useful to get an experimental result. Re-implementation could have been time-consuming, and it would be difficult to judge whether shortcomings are due to re-implementation vs intrinsic limitations of an algorithm. Overall, it would be important for the research community to devote more attention to enabling outside evaluation, even in cases where the source code can not be shared publicly.

## V. KEY INSIGHTS AND CONCLUSIONS

The high-level insights from our experimental analysis can be summarized as follows:
– A basic VAE with convolutional networks and a large batch size can outperform advanced approaches, including sequential autoencoders (SVAEs) that utilize LSTMs and structured variational inference with unsupervised disentanglement. This could be because non-stationary observations are challenging for advanced approaches, which require more gradient updates to significantly shift their posteriors (and might also need careful selection/decay of learning rates). Rather surprisingly, unsupervised disentanglement (e.g. $\beta$-VAE) performs worse (or at best similar to) the basic VAE.
– Predictive SVAE with a simple fully-connected architecture instead of an LSTM can outperform more advanced approaches. It can provide benefits for encoding velocity and contacts information into the latent state. This points to the opportunity for developing algorithms that interpolate between model-free and model-based RL. We could create latent representations that benefit from capturing the essence of an extended forward model $p(s_t, ..., s_{t+k}|s_{<t}, a_{<t})$ in the latent state, without explicitly mandating the use of a model-based RL algorithm.
– Approaches that introduce additional latent space structure can help faster learning, but can also impair ability to retain precise information about the underlying state. SPAIR [19], a faster version of AIR [20], mandates encoding explicit location variables. This helps it to roughly capture locations of all the

objects in the scene. However, for our newly proposed domains with objects of non-uniform sizes, the precision of locations does not improve with longer training. In contrast, a basic VAE completely fails to capture all but the largest objects in the scene, even if we simplify them to be simple single-color geometric shapes and remove the scene background. Nonetheless, VAE does learn to encode both position and orientation of the largest object in the scene much faster and more precisely than the structured SPAIR approach. Taking a broader view, we could postulate that for the field of robotics it is important to incorporate domain knowledge and structure into unsupervised learning methods. It might not be appropriate for roboticists to strive for removing all prior knowledge, even for approaches that aim for generality.

 – The most non-intuitive finding was that the stumbling block was not in the latent space learning, but in decoding. It is known that reconstructing small but salient parts of the scene can be challenging. Hence, in our experiments we enlarged the objects as much as possible. Thus, most objects were not small in terms of their pixel area. All of the basic decoder architectures we tried failed to reconstruct multi-object scenes. Supervised learning of the decoder with true low-dimensional state as input also did not produce high-quality reconstructions for the more advanced domains. For domains where decoding was tractable, our alignment analysis showed that the latent state tended to encode all the relevant information long before the decoder was able to produce discernible images. This points to a fundamental mismatch between the difficulty of encoding vs decoding. In turn, this points to the need to consider latent space learning that does not involve reconstruction. Despite excellent results for dataset-oriented applications, reconstruction might not be the best choice for settings with streaming non-stationary data, the kind we would like to handle in robotics.

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
