# OpenReview forum: "Benchmarking Unsupervised Representation Learning for Continuous Control"
_roboticsfoundation.org/RSS/2020/Workshop/RobRetro — RobRetro 2020_

### Official Review · AnonReviewer1 · 2020-06-20
**Nice comparison of different reconstruction based representation learning methods on diverse tasks**

**Rating:** 9
**Confidence:** 4

**Review:**

This paper presents a detailed comparison of several reconstruction based unsupervised representation learning methods for the task of continuous control. These methods are evaluated on several simulated datasets ranging from continuous control tasks such as the cartpole and ant to tasks involving object manipulation with a robot arm. The key metric for evaluating the quality of learned representations is their alignment to the true state (e.g. joint angles, object poses) — this is measured by training a NN to predict states from the latent representation.

Several reconstruction based learning methods are evaluated including single-step VAE and multi-step predictive VAE, a beta-VAE, Sequential VAEs and finally SPAIR. These are trained on data that is collected as an RL agent is training on the underlying environment and periodically evaluated. Several findings are reported including the relative strengths of a simple predictive VAE, the VAE_rpl and the potential failings of reconstruction based approaches in learning certain parts of the state well.

Overall, I found the discussion in this paper quite good. The baselines and evaluation tasks are well chosen and thoroughly evaluated and the results are quite interesting. A few key points:
1. It is not obvious from the paper whether the learned representations are used for training the RL agent on the task. I presume that this is not the case but it would be good to explicitly clarify this point.
2. Regarding the evaluation, is it computed on held out data or the same data that is currently in the replay buffer of the agent (which, incidentally is also the data the representation was trained on).
3. It would be interesting to see a baseline where the representation learning methods are trained purely offline, but on data that was collected with an RL agent throughout it’s training. Training offline in this data can be a loose upper bound on the performance of the underlying representation as the offline learner has access to a wider data distribution than the online learner where the data distribution is constantly changing. This could also explain why the VAE_rpl approach that uses random data in its replay buffer has such a good performance improvement in comparison to the standard VAE.
4. The paper presents several nice insights such as the difficulty of reconstruction based methods in learning a good representation. Particularly, I found the finding that “decoding as opposed to latent space learning was the main stumbling block” quite interesting. As the paper points out, reconstruction based methods do struggle on smaller details. Though in practice I have found that good structuring of a decoder, for example via the use of bilinear upsampling followed by convolutional layers, and an initial fully connected layer to get the lowermost level activations from the latent vector (as opposed to any form of tiling) to work well across many settings including real-world manipulation tasks. Much can also be gained by looking at the loss function for decoding, using a cross-entropy loss (even in regression as opposed to classification) gave slight improvements in reconstruction quality. It would be great if the authors could consider doing a sweep over different decoding setups in the context of these tasks (for a single learning algorithm, say VAE_rpl). This would be a valuable addition to the presented results.
5. Regarding the poor prediction of object orientation, it is well known that predicting orientation, especially for symmetric objects is hard. Regarding position prediction, the results are a bit surprising. I do have a question in this regard — what is the underlying task from which the data was collected? Was this a task where all the objects were being manipulated? If this is not the case then there is likely to be no data that has these objects moving — this could reduce the need for the representation to map the position of these objects (though they would vary between restarts).
6. Finally, as the authors mentioned, it would be interesting to explore losses that don’t rely on reconstruction. Is it possible to add a baseline based on a contrastive learning, such as the InfoNCE loss based representation learning?

---

> ### Author Response · Authors · 2020-07-04
> **response from the authors to the review**
>
> Thank you very much for your detailed comments. Below are our responses to each of the 6 points.
>
> 1. "not obvious from the paper whether the learned representations are used for training the RL agent on the task"
>
> We added the following clarification to the final version:
> To decouple unsupervised learner performance from RL training: for these experiments RL was trained on simulator state. Our evaluation suite supports training RL from the current latent representation. This is useful when analyzing one unsupervised method, but when comparing different methods this would likely cause RL learners to learn in different ways and at different rates. In turn, this could cause evaluation on pi_curr to refer to incomparable policies with vastly different success rates. For example: if RL gets stuck it might produce the same failed end state often, which would be easy to reconstruct due to low variability in frames, but would not constitute a success of the unsupervised learner.
>
> 2. "the evaluation, is it computed on held out data or the same data that is currently in the replay buffer of the agent"
>
> The evaluation is on held out data that is never added to replay buffers. Added the following clarification to the final version:
> Frames used for evaluation were held out from training i.e. not added to replay buffers at any time.
>
> 3. "would be interesting to see a baseline where the representation learning methods are trained purely offline"
>
> Right, an upper bound on performance might be interesting to consider for other applications. In our case we were specifically interested in how well the training would do with non-stationary frame distributions. That's why we did not investigate how well the methods would do in simpler learning settings.
>
> 4.  A few points were raised here, so addressing each in a separate paragraph
>
> Regarding "bilinear upsampling": it would be interesting to see pointers to specific architectures that do well. For example, from threads like this on pytorch forums it seems that a generic ConvTranspose2d layer would give flexibility, while a fixed upsampling strategy might be better for speed (https://discuss.pytorch.org/t/torch-nn-convtranspose2d-vs-torch-nn-upsample/30574/6). So it is not immediately obvious which option would ensure good performance (regardless of speed or flexibility) and would be good to see concrete examples.
>
> Regarding "an initial fully connected layer to get the lowermost level activations from the latent vector (as opposed to any form of tiling)":
> We tried an initial fully connected layer for the sequential VAE versions, but it did not seem to significantly improve performance. Perhaps it helps in combination with another technique (like a particular choice of upsampling strategy, as you point out).
>
> Regarding "It would be great if the authors could consider doing a sweep over different decoding setups in the context of these tasks":
> We might be able to spend a bit more resources to attempt this in the future. Though at the moment we decided to focus on exploring alternative unsupervised approaches that do not involve reconstruction, since reconstruction has other significant shortcomings (e.g. the need for additional losses to avoid ignoring small objects, etc).
>
> 5. "it is well known that predicting orientation, especially for symmetric objects is hard"
>
> We did not try to estimate origination for completely symmetrical objects like spheres, of course. But the orientation of the dominant axis (longest side of a box, axis of a tall cylinder, etc) should have been reasonably tractable.
>
> "what is the underlying task from which the data was collected? Was this a task where all the objects were being manipulated? If this is not the case then there is likely to be no data that has these objects moving"
>
> We tried a number of different setups: from random policies to RL with a goal to rearrange the objects to be in certain target positions. All of these were equally challenging, except for the cases where objects were static because the robot arm did not reach them. Of course this later cases was not interesting, since the distribution of frames was rather trivial then. We also tried to learn from a completely randomized version: no physics, but placing objects in random positions and orientations. We could not find a setting that would be non-trivial but easier for the unsupervised learners. Perhaps this could be discovered in the follow-up work.
>
> 6. "Is it possible to add a baseline based on a contrastive learning, such as the InfoNCE loss based representation learning?"
>
> Thank you for this suggestion. We also more recently got a related suggestion from another source to look further into contrastive methods. So indeed, this seems like an interesting direction. It would take a bit more effort to set up a comprehensive comparison, so we will plan to add such comparisons in the future.

---

### Decision · Program_Chairs · 2020-06-25

Accept